# MECD: Unlocking Multi-Event Causal Discovery in Video Reasoning

**Tieyuan Chen[1]\*, Huabin Liu[1]\*, Tianyao He[1], Yihang Chen[1], Chaofan Gan[1],**
**Xiao Ma[2], Cheng Zhong[2], Yang Zhang[2], Yingxue Wang[3], Hui Lin[3], Weiyao Lin[1]†**
[1] Shanghai Jiao Tong University, [2] Lenovo Research, AI Lab,
[3] China Academic of Electronics and Information Technology
{tieyuanchen, huabinliu, wylin}@sjtu.edu.cn
https://github.com/tychen-SJTU/MECD-Benchmark

## Abstract

Video causal reasoning aims to achieve a high-level understanding of video content from a causal perspective. However, current video reasoning tasks are limited in scope, primarily executed in a question-answering paradigm and focusing on short videos containing only a single event and simple causal relationships, lacking comprehensive and structured causality analysis for videos with multiple events. To fill this gap, we introduce a new task and dataset, **M**ulti-**E**vent **C**ausal **D**iscovery (MECD). It aims to uncover the causal relationships between events distributed chronologically across long videos. Given visual segments and textual descriptions of events, MECD requires identifying the causal associations between these events to derive a comprehensive, structured event-level video causal diagram explaining why and how the final result event occurred. To address MECD, we devise a novel framework inspired by the Granger Causality method, using an efficient mask-based event prediction model to perform an *Event Granger Test*, which estimates causality by comparing the predicted result event when premise events are masked versus unmasked. Furthermore, we integrate causal inference techniques such as front-door adjustment and counterfactual inference to address challenges in MECD like causality confounding and illusory causality. Experiments validate the effectiveness of our framework in providing causal relationships in multi-event videos, outperforming GPT-4o and VideoLLaVA by 5.7% and 4.1%, respectively.

## 1  Introduction

Video causal reasoning aims to achieve a high-level understanding and analysis of video content from a causal perspective. Video Question Answering (VQA) [1–5] represents one of the most prominent tasks in causal reasoning, where models are tested on their causal ability to understand video content through causal questions such as explanations, predictions, and counterfactual assumptions. Recently, some studies have sought to move beyond the single QA task, attempting to construct more complex and challenging video reasoning tasks and methodologies. For example, CLEVRER [5], V-CDN [6] and CATER [7] explored more difficult causal reasoning tasks in virtual scenes by constructing object-aware features or using graph neural networks. Neural-symbolic paradigm AAR [8] and LMLN [9] extended to derive inference rules by symbolizing data. VAR [10] and BiGED [11] extended to daily video causal reasoning by introducing causality during prediction.

However, current video causal reasoning tasks are still limited in scope (primarily QA-based) and mainly focus on short videos containing only a single event or a few events. Most importantly, they

---

\*Equal Contribution.
†Corresponding Author.

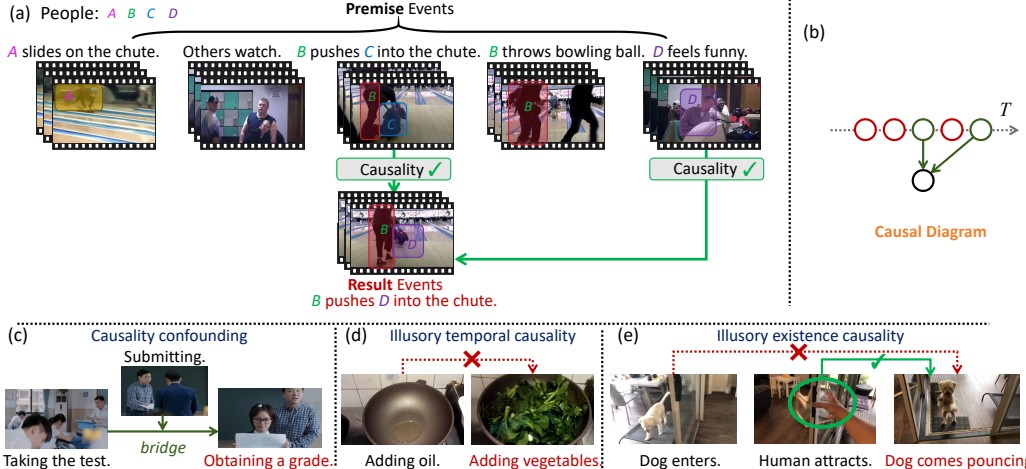

Figure 1: (a): Illustration of Multi-Event Causal Discovery Task, where the 3rd and 5th premise events account for the occurrence of the final event. The objective of our task is to determine whether a causal relation exists between events and outputs a structured causal diagram. (c): Example of causality confounding. (d)&(e): Illustration of illusory causality.

cannot provide a comprehensive and structured causal representation for multi-event video reasoning, which is typically required in real-world scenarios. For instance, in traffic surveillance videos, it is necessary to cross-analyze events happening at different times to determine which events, or combinations of events, led to the final traffic accident event.

To address this gap, we set up a new task: **M**ulti-**E**vent **C**ausal **D**iscovery (MECD), which aims to uncover causal relationships among events that distribute chronologically in long videos. As illustrated in Fig. 1, given multiple chronologically arranged event segments in a video along with their corresponding textual descriptions (Fig. 1(a)), MECD requires identifying causal associations between these events to derive a comprehensive and structured event-level causal diagram (Fig. 1(b)), indicating why and how the final result event happens. Meanwhile, we contribute a new dataset for the training and evaluation of MECD by collecting long-form videos involving multiple events and manually annotating real causal relations between events for them. However, to our knowledge, no available solutions can directly comprehend causal relationships at the event level, necessitating the development of a new framework to tackle this complex task.

To this end, we draw inspiration from the *Granger Causality Method* [12–14] for solution, which is widely used in traditional causal discovery for low-dimensional time-series data (e.g., stock prices, weather patterns). The main idea is that temporal causality can be effectively estimated by predictive ability. Specifically, applied to videos, if Event A occurs prior to Event B, we consider A to be a cause of B only if A could facilitate the prediction of B. We term this criterion the *Event Causality Test*. However, compared to simple low-dimensional data, the inputs of MECD involve much more complex modalities, including both visual and textual content, which may introduce bias in the estimation of causality using such a predictive paradigm. Specifically, we observe that directly applying *Event Causality Test* to video causal discovery presents two main problems:

(1) **Causality confounding** indicates that the original causal relationships between events are disrupted or interfered with by other relay or adjacent events. Such confounding stems from the fact that many causal relationships flow through an intermediary event that acts as a bridge. As shown in Fig. 1(c), event "submitting the paper" serves as a necessary bridge between "taking the test" and "obtaining a grade." In this case, this bridge event might be mistakenly regarded as the only cause of the result event, while another cause, "taking the test," is overlooked. However, the bridge event can only occur if "taking the test" happens first. Therefore, we cannot identify the real causality between events that linked by such bridges following a simple predictive criterion, and eliminating such confounding is thus crucial for an accurate causal discovery.

(2) **Illusory Causality**, which includes illusory temporal and existence causality. Illusory temporal causality exists when events exhibit a close correlation in temporal distribution. Such correlation may mislead the test of real causality. As shown in Fig. 1(d), the event "adding oil

when cooking" often occurs before "adding vegetables to stir-fry," but there is no real causality between them. As for *illusory existence causality*, it occurs when some objects in early events may serve as necessary existence conditions of a later event. For instance (Fig. 1(e)), consider determining the causal relation between "a large brown dog enters the room" (at the start of the video) and "the dog runs towards the camera." (at the end of the video). Although the presence of the dog in the former event is a prerequisite for the subsequent event, it does not directly cause the dog to rush towards the camera.

Building upon the preceding discussion, we introduce a novel framework to tackle MECD. This framework executes the *Event Granger Test* via an efficient mask-based event prediction model. It deduces the causality of a premise event by comparing the predicted features of the result event when the premise is either masked or unmasked. Furthermore, to mitigate the challenges of causality confounding and illusory causality discussed earlier, we integrate two additional causal inference techniques—front-door adjustment [15–17] and counterfactual inference [15, 18, 19]—into our framework. Specifically, these techniques compensate for or remove the causal effects of previous or subsequent adjacent bridge events to eliminate confounding. Simultaneously, they address the issue of illusory causality through the incorporation of an extra chain of thought [20–22] and existence-only descriptions during inference. Extensive experiments validate the effectiveness of our proposed framework in predicting structured causal relationships for given long-form videos.

## 2 Related Work

**Video causal reasoning** Many tasks in the past have tried to carry out causal reasoning in videos. Among these, the most common task is Video Question Answering (VQA) [1–4], aiming to give a reasonable answer to the question, methods such as SeViLA and LocAns [3, 4] made abductions based on the result, they grounded a single reason in previous time. However, VQA does not extend to abduct multiple reasons, merely creating a single causal link from reason to result.

Many tasks were based on VQA task for further causal reasoning attempts. CLEVRER [5], CATER [7] and V-CDN [6] explored causal reasoning based on physics and other basic laws in virtual scenes. However, these tasks haven't been committed to extending to the general video causal reasoning. AAR [8] and LMLN [9] symbolized data and derived inference rules using external knowledge. However, they can only reason within a defined symbol domain. The most similar VAR [10] predicted explanation events with premise events, and the causality was introduced during its prediction process. However, firstly it hasn't been committed to discovering the complete causal diagram. Besides, there is no explicit utilization of causal methods which constrains its ability.

All tasks above are for causal reasoning in short videos, while ours aims to handle long-duration videos. Besides, most of these are coarse video-level tasks, ours is more fine-grained event-level reasoning. Additionally, we want to establish a whole causal diagram rather than a single causal link. In conclusion, all these tasks haven't been committed to discovering causality among complex multi-event videos. Consequently, there exists a necessity need for a more comprehensive task.

**Causal discovery in low-dimensional temporal data** Traditional causal discovery methods of simple temporal data are mainly divided into three categories. Constraint-based methods use conditional independence tests to identify causal relations [23–25]. Score-based methods search through the space of all possible causal structures to optimize a specified metric [26–28]. The Granger Causality method discovers causal relations by calculating the degree to which the earlier occurred event contributes to the prediction of the latter occurred event [29–31]. The constraint-based and score-based methods require stringent assumptions about data distribution, making them less suitable for video data. The Granger Causality methods are more suitable yet face challenges when applied directly to video data, our method reaches better performance by utilizing causal inference methods.

## 3 Benchmark

### 3.1 MECD task settings

Our Multi-Event Causal Discovery (MECD) task is designed to test the ability of causal discovery in multi-event videos. Given a video $\mathcal{E}$ that contains chronologically organized $N$ events, $\mathbb{E} := \{e_1, \ldots, e_N\}$, the task aims at determining whether any previous event $e_n$ $(n < N)$ has a causal

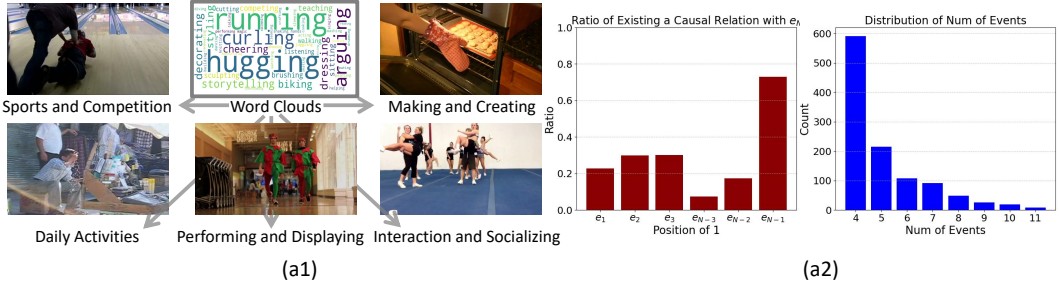

Figure 2: **Constitute of MECD dataset.** In (a1), we present 5 main video categories of the dataset. The word cloud is also summarized for video types. In (a2), the left chart indicates the impact of positions of events on their causality where we find the second last event tends to be more significant; while the right chart plots the number of events in videos.

relation with the last one (*i.e.*, $e_N$). Specifically, an event $e_n = \{v_n, c_n\}$ consists of a video clip $v_n$ and the corresponding caption $c_n$. Without loss of generality, relations of previous events to the last one can be expressed as $\boldsymbol{r} = [r_1, \ldots, r_{N-1}]$, where $r_k$ $(k < N)$ is set to "1" to indicate the existence of $e_k$'s causal relation with $e_N$, and "0" otherwise. Notably, this setting is generalizable to causal relations of any of two events as long as we cut off the video and treat the latter one as the last event.

## 3.2 MECD task dataset

**Data Source** The Multi Events Causal Discovery (MECD) task contains videos with multiple events and intricate causal relationships. The ActivityNet Captions dataset [32] is built on ActivityNet v1.3 which includes 20k 120-second YouTube untrimmed videos. We carefully reorganize the ActivityNet Captions dataset and select 1,105 lifestyle videos that span diverse scenarios. We call this new dataset as MECD dataset, where 806 and 299 videos are randomly split for training and testing, respectively. Specifically, each video in the MECD dataset contains 4 to 11 events, with a minimum of 2 premise events exhibiting causal relations with the last one. Fig. 2 (a1) presents the main categories and word clouds of video types. Please refer to Appendix Sec. B.4 for more dataset examples.

**Data Cleaning** We further clean our dataset by excluding non-causal videos. For example, videos that describe multiple non-causal action steps such as washing hands and shaving were excluded.

**Dataset Annotation** The annotations of MECD dataset include 4 attributes. The "duration", "sentence", and "timestamps" attributes in annotations remain the same as the ActivityNet Captions dataset. Specifically, in the context of our task, a new attribute "relation" is introduced. To obtain this attribute, relations among events are firstly annotated by GPT-4 API [33], and subsequently refined by five human annotators. Through a cross-annotation process, gt labels are determined by the majority of the annotators' causal relation choices, thus mitigating potential inaccuracies and subjective biases to a certain extent. We also present the impact of positions of events on their causality and number of events in videos in Fig. 2 (a2), annotation pipeline is in Appendix Sec. B.3.

## 4 Methodology

In this section, we present our Video Granger Causality Model (VGCM), as shown in Fig. 3. This model establishes the global connections across all events, and deduces the causality of a premise event by comparing the output features when it is masked or not, under the concept of the *Event Causality Test*. However, masking out an event may lead to the problem of confounding and illusion. In this context, we further utilize causal inference methods to address these by compensating or removing the effect of previous or subsequent causal events to mitigate the confounding meanwhile during inference the extra chain of thoughts and existence-only descriptions relieve the illusion.

### 4.1 VGCM: Video Granger Causality Model

Building upon the Granger Causality method introduced in [34–36], our core motivation for constructing VGCM is *Event Causality Test*: To compare the prediction result of the last event using all the premise events with or without a certain event in it. If the results exhibit obvious divergence, it indicates that the current premise event is causally related to the result event.

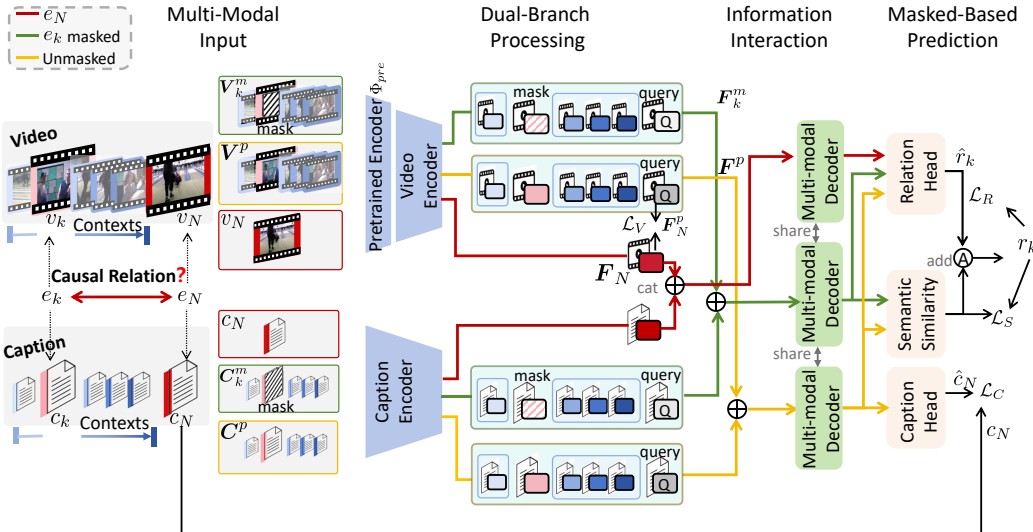

Figure 3: **Video Granger Causality Model.** Two data streams $V^p$ and $V_k^m$ serve as input, video and text embeddings are concatenated after being separately embedded. The VGCM incorporates two classifiers, the caption head takes the unmasked stream to accomplish the event-predicting task, while the relation head discovers the causal relations with two embedding streams.

We design VGCM to take in both the video clips and the captions to maximize information utilization. As illustrated in Fig. 3, our proposed VGCM is a multi-modal transformer-based structure with a video encoder and caption encoder, and a multi-modal decoder with causal relation head to discover causal relations through the predicting process and the comparison of predicting results.

Based on this, we denote $\mathbb{E}^p$ as the set of all the *premise* events $\mathbb{E}^p := \mathbb{E} \setminus e_N$, and $\mathbb{E}_k^m := \mathbb{E}^p \setminus e_k$ as the event set where the premise event $e_k$ ($k < N$) is masked. Notably, we mask the event $e_k$ by setting all zeros to its video clip $v_k$ and assign constant characters to the caption $c_k$.

Following [10, 37–39], we firstly pretrain a video encoder $\Phi_{pre}$ under an action recognition task to extract the features of the video clips. We essentially create two paths, one for the unmasked event set $\mathbb{E}^p$ (orange path in Fig. 3) while the other for the set with one event (*i.e.*, $e_k$) masked $\mathbb{E}_k^m$ (green path in Fig. 3). The video clips and captions are first separately encoded using $\text{Enc}_V$ and $\text{Enc}_C$ to obtain compact features, then their features are sent to a multi-modal decoder Dec that shares weights for both paths to fuse the information. Afterward, several model heads are employed for feature comparison and loss measurement. $V^p$ and $C^p$ are the video clip and caption matrix concatenated from all premise events set $\mathbb{E}^p$, similarly, $V_k^m$ and $C_k^m$ are from $\mathbb{E}_k^m$.

$$F^p = \text{Enc}_V(\Phi_{pre}(V^p)), \quad O^p = [\text{Dec}(\text{Cat}(F^p, \text{Enc}_C(C^p)))]_{N-1}$$
$$F_k^m = \text{Enc}_V(\Phi_{pre}(V_k^m)), \quad O_k^m = [\text{Dec}(\text{Cat}(F_k^m, \text{Enc}_C(C_k^m)))]_{N-1} \quad (1)$$

where $\text{Enc}_V$ and $\text{Enc}_T$ represent the encoder module of video clips and captions, respectively. Dec is a multi-modal decoder that shares weights for both paths. Cat indicates the concatenate operation, and $[-]_{N-1}$ indicates the $(N-1)$-th slice at dimension 0. $F^p$ and $F_k^m$ are features after encoding, and $O^p$ and $O_k^m$ are the output features, which are then used for comparison of difference. Incorporating both visual and linguistic representations, the decoder conducts cross-modal reasoning and leverages the context from the unmasked premise events to posit a meaningful representation of the most likely explanatory result event.

Subsequently, the feature $O^p$ deduced from the unmasked events is sent to the caption head for prediction. Additionally, in order to compare the difference of the prediction result, $O^p, O_k^m$ are directed to the relation head for causal relation discovery. The result event $e_N$ is encoded the same way as $e_k$ to get feature $F_N = \text{Enc}_V(\Phi_{pre}(v_N))$ and the output $O_N = \text{Dec}(\text{Cat}(F_N, \text{Enc}_C(C_N)))$, $O_N$ is aggregated for reasoning (red path in Fig. 3). The relation head consists of a semantic query module and a self-enhancement module, where outputs are concatenated and then passed through the cross-reasoning layer $g_r$ for further interaction. Last but not least, the auxiliary similarity is measured between $O^p$ and $O_k^m$ as a supplement to the output information of the relation head. After

the reasoning process, the prediction output of the causal relation $\hat{r}_k$ can be represented by:

$$\hat{r}_k = g_r(\text{Cat}(\Phi_{att}^C(\text{Cat}(\boldsymbol{O}_k^m, \boldsymbol{O}_N), \text{Cat}(\boldsymbol{O}^p, \boldsymbol{O}_N)), \Phi_{att}^I(\text{Cat}(\boldsymbol{O}_k^m, \boldsymbol{O}_N)))) \quad (2)$$

where $\Phi_{att}^C$ represents cross-attention, $\Phi_{att}^I$ represents self-attention, $g_r$ represents linear layer. The training objective consists of two main directions as previously discussed:

To reconstruct the textual and visual representation of the result event $e_N$, we introduce caption loss and reconstruction loss, respectively. Caption loss $\mathcal{L}_C$ ensures an accurate prediction of the result caption $\hat{c}_N$ given all the premise events $\mathbb{E}^p$. Simultaneously, visual reconstruction loss $\mathcal{L}_V$ forces the encoder to "imagine" a representation of the result video clip $\hat{v}_N$ that better aligns with the original representation $v_N$. These losses allow the model to predict visual and textual representations that are close to the original representations, which better supports our method of inferring causal relations by comparing the results of the two-stream predictions.

For the objective of causal discovery, we introduce causal relation loss and an auxiliary semantics similarity loss. Causal relation loss $\mathcal{L}_R$ supervised the output relations $\hat{r}_k$. Meanwhile, the semantics similarity loss $\mathcal{L}_S$ is introduced to guarantee the semantics similarity of result event prediction under the presence or absence of a causal-relation-free premise event. The complete loss function is:

$$\mathcal{L} = \mathcal{L}_C(c_N, \hat{c}_N) + \lambda_R \mathcal{L}_R(r_k, \hat{r}_k) + \lambda_V \mathcal{L}_V(\boldsymbol{F}_N^p, \boldsymbol{F}_N) + \lambda_S \text{sign}(r_k)\mathcal{L}_S(\boldsymbol{O}_k^m, \boldsymbol{O}_p) \quad (3)$$

where $\lambda_R$, $\lambda_V$, and $\lambda_S$ are weights for trade off. $\mathcal{L}_C$ and $\mathcal{L}_R$ are the cross-entropy losses, $\mathcal{L}_V$ and $\mathcal{L}_S$ are the mse losses, $\boldsymbol{F}_N^p$ is the N-th slice of $\boldsymbol{F}^p$, which represents the encoded feature of $e_N$.

## 4.2 Causal Inference in VGCM

In Sec. 4.1, we employ the concept of Granger Causality to design our VGCM model under the principle of *Event Causality Test* which may, however, introduce causality confounding and illusory. Below we introduce these issues in detail, as well as how we manage to solve the problems.

**Causality confounding** is a phenomenon where the original causal relations across events are impacted due to modification (*i.e.*, masking) of some intermediate events (*i.e.*, $e_k$). Existing disentangled representation learning works [40, 41] disentangled different attributes of a variable by supervising high-order distribution under strict assumptions but failed in disentangling different variables.

Specifically, when $e_k$ is masked for the comparison in VGCM, the causal relations between $e_k$'s adjacent events and the last event are impacted, leading to a confounding of causal effects. Notably, for brevity, we only employ $e_k$'s previous one event $e_{k-1}$ and its subsequent one event $e_{k+1}$ for analysis, but the same analysis also applies to all the previous or subsequent events. To be specific, there exist two distinct kinds of confounding when $e_k$ is absent: **1)** Causal effects of $e_{k-1}$ to $e_N$ may be lost, as its connection to $e_N$ is built upon $e_k$, (green path in Fig. 4 (a1)). **2)** Causal effects of $e_{k+1}$ to $e_N$ may be redundant, as $e_k$ must be a necessary prior of its causality, (red path in Fig. 4 (a1)).

**Illusory causality** is another issue that may lead to some spatial or temporal misunderstandings, including illusory temporal and existence causality. **1)** Illusory temporal causality is the situation that events could have tight temporal ordering, but they in fact have no causal relations. **2)** Additionally, illusory existence causality occurs when an object introduced in the premise event is a necessary condition for the result, but the premise event does not semantically serve as a reason. Notably, we find that illusory in multi-event videos is much more significant than two independent events, which also tends to be exacerbated by causality confounding.

Overall, **causality confounding** and **illusory causality** both bring difficulties for relation modeling of events in videos. Notably, these two issues are coupled in that **causality confounding** tends to exacerbate **illusory causality** by misallocating attention to temporal ordering and causal effect. Therefore, illusory causality can be partially relieved by solving the problem of causality confounding.

When considering taking the illusory causality, the chain of thoughts [20–22] has been shown in LLMs to lead the model to logical thinking which is similar to human thought process, the chain of thoughts $T_{cot[e_{k-1}:e_N]}$ provides a step-by-step process of reasoning the $e_N$ from $e_{k-1}$. Specifically, $T_{cot[e_{k-1}:e_N]}$ is obtained using GPT-4 API [33] by feeding it with $e_{k-1}$, $e_N$ along with a prompt asking it to provide the probable reasoning chain. We consider utilizing it in causal inference to eliminate the attention bias on temporal correlations introduced by non-causal temporal knowledge.

Besides, as the illusory existence causality is caused by the objects' correlation between the events, we address this influence by keeping objects in the green path in Fig. 3 the same as those in the

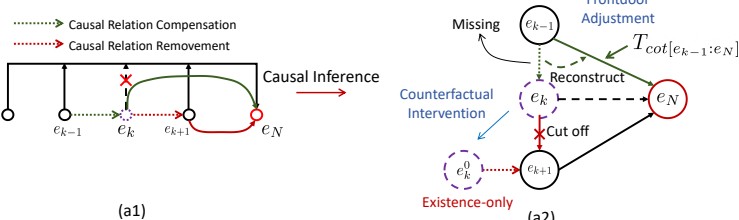

Figure 4: **Causal Effect of the Adjacent Events and Causality Diagram.** (a1) shows the causality of the third event analyzed, the red causal effect needs to be compensated while the green needs to be mitigated. (a2) shows the causal inference methods corresponding to the two causal effects.

orange path. We introduce an alternative event $e_k^0 = \{v_k^0, c_k^0\}$ of $e_k$ to briefly recaps the objects in $e_k$. Specifically, $c_k^0$ is obtained using GPT-4 API [33] by feeding it with $c_k$ along with a prompt asking it to extract the objects from $c_k$ and organize them as the sentence such as "There are objects A, B and C.". Consequently, we opt to employ $c_k^0$ to approximate $e_k^0$ in our VGCM model while omitting $v_k^0$, as $c_k^0$ is sufficient already to convey the information of objects. By providing $e_k^0$, all the necessary objects are still available in this path, thus effectively mitigating the illusory existence causality, facilitating the model to focus more on essential and causality-related semantic information.

To tackle the issues above, we introduce two causal inference methods: the front-door adjustment [42] for the missing causal effect of $e_{k-1}$ and counterfactual inference [42] for the redundant causal effect of $e_{k+1}$. Meanwhile, the chain of thoughts $T_{cot[e_{k-1}:e_N]}$ and the descriptions of existence $c_k^0$ are also provided to carefully address illusory causality, which in turn mitigates confounding.

We establish a causality diagram in Fig. 4 (a2) for an improved elaboration. On masking $e_k$, the causality confounding that requires compensation $\boldsymbol{F}^C$ or removement $\boldsymbol{F}^R$ can be expressed as:

$$\boldsymbol{F}^C = P(e_N|e_k) - P(e_N|do(e_k)), \boldsymbol{F}^R = P(e_N|e_{k+1}) - P(e_N|do(e_{k+1})) \qquad (4)$$

where $P(e_N|e_k)$ and $P(e_N|e_{k+1})$ represents the process by which we predict $e_N$ from $e_k$ and $e_{k+1}$ in the orange path in Fig. 3, and $do(\cdot)$ represents do-operation in causal inference [15] that cuts off the causal relation between the event and its causes.

We aggregate the subsequent events $e_{k+1}$, the current event $e_k$ and the chain of thoughts $T_{cot[e_{k-1}:e_N]}$ using a linear layer $g_{do}$ for aggregation and the cross-attention and self-attention, according to the study in [43, 17], $P(e_N|do(e_k))$ can be implemented as:

$$P(e_N|do(e_k)) = g_{do}((\text{Cat}(\Phi_{att}^C(e_k, e_{k+1}, e_{k+1}), \Phi_{att}^I(e_k, e_k, e_k), \text{Enc}_c(T_{cot[e_{k-1}:e_N]})))), \qquad (5)$$

Here, we re-use the cross-attention $\Phi_{att}^C$ and the self-attention $\Phi_{att}^I$ modules as in (2) to cut off the causal effect from $e_{k-1}$ to $e_k$ through do-operation, $e_k$ only interacts with subsequent events in predicting $e_N$. Then the missing causal effect $\boldsymbol{F}^C$ can be compensated since the causal-view operation and illusory temporal causality can be suppressed at the same time with the introduction of the chain of thoughts. Similarly, the redundant causal effect $\boldsymbol{F}^R$ can be removed by applying counterfactual intervention, then $P(e_N|do(e_{k+1}))$ can be represented by:

$$P(e_N|do(e_{k+1})) = P(e_N|e_{k+1})[P(e_{k+1}|e_k) - P(e_{k+1}|e_k^0)], \qquad (6)$$

$P(e_N|do(e_{k+1}))$ effectively cuts off the redundant causal effect between $e_{k+1}$ and $e_N$ for the reason that the causes of $e_{k+1}$ are replaced with counterfactual description $e_k^0$, then the illusory existence causality can be suppressed at the same time.

To refine the originally decoded feature $\boldsymbol{O}_k^m$ from the path with premise events masking:

$$\boldsymbol{O}_k'^m = \boldsymbol{O}_k^m - \text{Dec}(\boldsymbol{F}^C) + \text{Dec}(\boldsymbol{F}^R) \qquad (7)$$

where $\boldsymbol{O}_k'^m$ is the refined feature that replaces $\boldsymbol{O}_k^m$ for further deduction of the model. With the refinement feature $\boldsymbol{O}_k'^m$, our VGCM model effectively compensates the connections between $e_{k-1}$ and $e_N$ that were originally lost due to the removal of $e_k$, and effectively removes the redundant causal effect between $e_{k+1}$ and $e_N$ as well.

Table 1: **Main results.** Experiments validate the effectiveness of our VGCM framework in reasoning causal relations towards multi-event videos, outperforming GPT-4o and VideoLLaVA by 5.7% and 4.1%, respectively. ‡ indicates without causal inference. Random results and human performances are also provided.

| | Paradigm | Method | Ave SHD ↓ | Accuracy ↑ |
|---|---|---|---|---|
| - | Random Guess | Guess all causal. | 6.95 | 42.4 |
| | | Guess all non-causal. | 5.36 | 57.6 |
| Few-shot | LLM Base | Gemini-1.5-Pro [44] | 4.91 | 59.3 |
| | | GPT-4-0613 [33] | 4.92 | 59.6 |
| | VLLM Base | MiniGPT4-video [45] | 5.16 | 56.8 |
| | | MiniGPT-4 [46] | 5.14 | 57.5 |
| | | Video-llama [47] | 5.10 | 60.6 |
| | | VideoChat2 [48] | 4.89 | 60.7 |
| | | VideoLLaVA [49] | 4.85 | 62.5 |
| | | GPT-4o [33] | **4.69** | **65.5** |
| Fine-tuned | Multi-modal | VAR [10] | 4.96 | 57.3 |
| | | Videobert [50] | 4.95 | 60.9 |
| | | CLIP (ViT-L/14) [51] | 4.77 | 62.9 |
| | VLLM Base | VideoChat2 [48] | 4.77 | 66.9 |
| | | VideoLLaVA [49] | **4.73** | **67.1** |
| | Ours | VGCM‡ | 4.51 | 67.0 |
| | | **VGCM** | **4.19** | **71.2** |
| - | Humans | Deductive Reasoning | 2.05 | 87.2 |

Table 2: **Ablation Study.** Adj indicates the front-door adjustment, and inter indicates the counterfactual intervention.

| Base designs | | | Causal methods | | Acc |
|---|---|---|---|---|---|
| $\mathcal{L}_C$ | $\mathcal{L}_V$ | $\mathcal{L}_S$ | Adj | Inter | |
| | ✓ | ✓ | | | 64.8 |
| ✓ | | ✓ | | | 65.1 |
| ✓ | ✓ | | | | 65.3 |
| ✓ | ✓ | ✓ | | | 67.0 |
| ✓ | ✓ | ✓ | ✓ | | 68.7 |
| ✓ | ✓ | ✓ | | ✓ | 69.3 |
| ✓ | ✓ | ✓ | ✓ | ✓ | 71.2 |

Table 3: Illusory existence causality experiment. w/o C indicates without counterfactual intervention.

| Method | starting division | Ending division |
|---|---|---|
| VGCM (w/o C) | 1.12 | 1.04 |
| **VGCM** | 1.12 | **0.93** |

Table 4: Illusory temporal causality experiment. w/o F indicates without front-door adjustment and Ave indicates average.

| Method | $r_0$ Acc | $r_{N-1}$ Acc | Ave $r$ Acc |
|---|---|---|---|
| VAR [10] | 53.8 (-3.5) | 54.6 (-3.7) | 57.3 |
| VGCM (w/o F) | 63.6 (-3.3) | 63.7 (-3.2) | 66.9 |
| **VGCM** | 68.0 (**-0.7**) | 68.4 (**-0.3**) | 68.7 |

# 5 Experiments

## 5.1 Main results

**Implementation details.** including the pretraining process, detailed architecture of VGCM, and hyper-parameters settings can be found in Appendix Sec. A due to space constraints.

**Baselines.** We mainly compared our model with basic multi-modal models such as baseline model Videobert [50] and widely used CLIP-L [51] and the most similar reasoning model VAR [10]. Besides, we also conduct experiments on powerful LLM, including GPT-4 [33] and Gemini-Pro [44]. VLLM utilized for comparison includes widely accepted GPT4-o [33], VideoLLaVA [49], MiniGPT-4 [46], Video-llama [47], VideoChat2 [48] and MiniGPT4-video [45]. Specifically, LLMs and VLLMs are conducted under the few shot setting (In-Context Learning) following the causal discovery tasks in NLP [52–54], additionally, we reported the performance of fine-tuned VideoLLaVA and VideoChat2.

**Metrics.** We utilize the top-1 accuracy of the output causal relation chains with respect to the final event to evaluate the model's capability in causal discovery. Although our VGCM is designed to discover the causal relations leading to the final event, when truncating the video during inference and redefining the final event as the new result, VGCM can generate a comprehensive causal diagram for the entire video without introducing additional training. Consequently, in addition to the primary metric accuracy, we introduce Structural Hamming Distance (SHD) [55, 56] as a supplementary metric. SHD measures the degree of matching between comprehensive causal graphs by summing the number of incorrect causal relations. In the MECD test set, the average number of causal relations in video causal graphs is 12.31, and a lower Ave SHD value of the test set indicates better performance.

**Results.** We report the quantitative results in Tab. 1. Our VGCM without causal inference reaches an accuracy of 66.9%, demonstrating basic reasoning capabilities. Furthery, the complete VGCM reaches a better performance with an accuracy of 71.2%, outperforming the GPT-4, GPT4-o, fine-tuned VideoLLaVA [49] by 11.6%, 5.7%, and 4.1%. Additionally, we explored the effect of altering the input format of the two modalities in Appendix Sec. C.1, indicating that VGCM is not dependent on the input format. The results compared with the two metrics indicate that for most models, accuracy is already adequate to represent their causal discovery capabilities. However, the additional metric Ave SHD indicates that Gemini and GPT-4 exhibit a superior overall capacity for discovering complete relations. An example of the output complete causal diagram is visualized in Figure 7.

GPT-4 [33] stands out as one of the most advanced LLM models, however, we found that even being provided with sufficient few-shot examples (detailed in Appendix Sec. C.2), its accuracy remains at

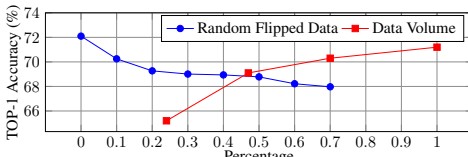

Figure 5: **Dataset robustness.** Accuracy decreases slightly when increasing noise, and increases slowly when increasing the training data.

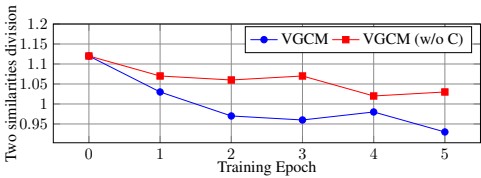

Figure 6: **Causality discovered analysis.** The similarity of masking causal premise events is obviously lower through counterfactual intervention.

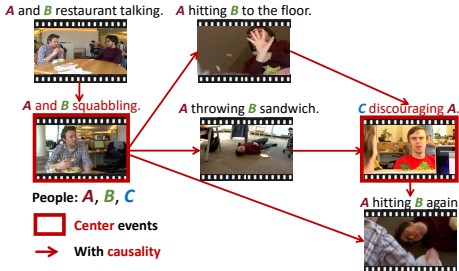

Figure 7: Complete causal diagram.

Table 5: Open-set ability of VGCM.

| Method | TOP-1 Accuracy |
| --- | --- |
| VAR [10] | 54.8 |
| VGCM[‡] | 59.2 |
| **VGCM** | **64.4** |

only 59.6%. Possible explanations may be due to task contamination [57], GPT-4 mainly performs well on datasets released before the training date, while our task is novel. Moreover, other reasons may include the causal hallucination problem of establishing a threshold for differentiating between scenarios with and without causality [58]. For further insights into GPT-4's failure cases, refer to Appendix Sec. B.2.

As illustrated in Tab. 6, we have assessed the inference speed of various models, with our VGCM achieving a swift 0.76 seconds per sample. The proposed method incurs an overhead of only 8.57% over the Videobert baseline. It is noteworthy that our inference speed is 3 to 6 times faster than that of all Video LLMs. The inference speed experiments were conducted on 1 NVIDIA A6000 GPU.

## 5.2 Ablation Study

**Video Granger Causality Model design** We designed our causal discovery model based on the Granger Causality, three auxiliary losses are applied. The performances in Tab. 2 indicate that our VGCM benefits from the design of $\mathcal{L}_V$ and $\mathcal{L}_C$, for they support our method of inferring causal relations by facilitating the model with event prediction ability. $\mathcal{L}_S$ also benefits our model by supervising the causal feature similarity of $e_N$ with and without non-causal event $e_k$ masked.

**Front-door adjustment with chain of thoughts candidate** The method does improve reasoning ability in Tab. 2. We conduct an experiment in Tab. 4 for further proof. Since events closer to the result event are higher as the cause, the model likely learns these biased time-domain tendencies. So we compare the accuracy of VGCM without front-door adjustment with chain of thought candidate and VGCM in determining the first relation $r_1$ and the last relation $r_{N-1}$. The results demonstrate that temporal illusory causality is greatly mitigated, visualization can be found in Fig. 8 Example 1.

**Counterfactual intervention with existence-only descriptions** The performance in Tab. 2 shows that counterfactual intervention with existence-only descriptions does facilitate the model with powerful reasoning ability. We dive into further analysis on the basis that when a non-causal event is masked, the causal feature $\boldsymbol{F}_k^m$ fed into the causal relation head should be similar to the unmasked feature $\boldsymbol{F}^p$, instead, a bigger gap appears when masking a causal event. For stronger proof, we measure the difference in feature similarity in Tab. 3 and Fig. 6. We define the similarities division as the quotient of the similarity($\boldsymbol{F}_k^m$, $\boldsymbol{F}^p$) with a non-causal $e_k$ masked over with a causal $e_k$ masked. In the experiment, we find that the similarity division is always above 1 without the counterfactual intervention, however, the existence illusory is solved with counterfactual intervention for the reason that the division is below 1 of VGCM, example visualization can be found in Fig. 8 Example 2.

## 5.3 Robustness Analysis

**Model Robustness** To prove our model's robust reasoning ability, we split the MECD dataset into five categories, and conduct an experiment similar to the open-set setting with cross-validation. VGCM

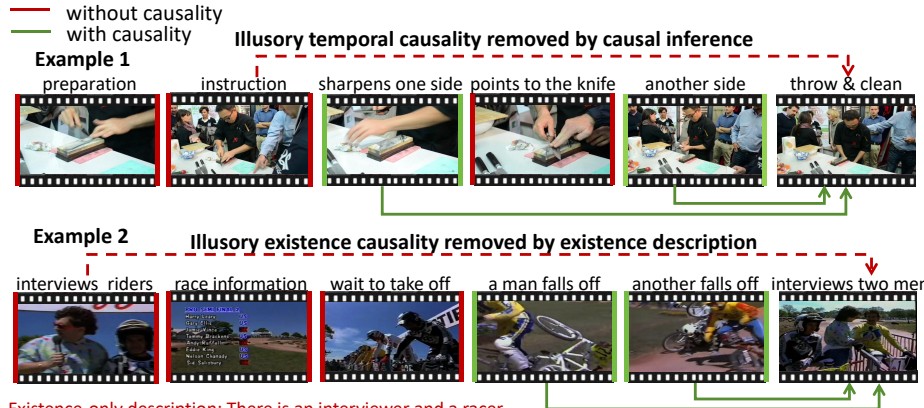

Figure 8: **Successful abduction examples of our VGCM.** Results indicate that after utilizing causal inference methods, illusory causality is suppressed and robust abduction ability is facilitated.

Table 6: **Infernce speed.** Our VGCM is 3-6 times faster than all Video LLMs while slightly slower than the baseline.

| Model | Inference Speed |
|---|---|
| Videobert [50] | 0.70 |
| **Our VGCM** | **0.76** |
| VideoLLaVA [49] | 2.12 |
| VideoChat2 [48] | 2.96 |
| MiniGPT4-video [45] | 3.98 |
| MiniGPT-4 [46] | 4.72 |

Table 7: **Model's generalizability test.** Higher VQA Acc and VQA Score are reached when prompted with causal relations from our VGCM.

| Output Causal Relations | VQA Acc | VQA Score |
|---|---|---|
| w/o (Standard QA setting for VLLMs) | 43.17 | 2.82 |
| w Gemini-Pro [44] | 49.10 | 2.90 |
| w GPT-4 [33] | 49.36 | 2.89 |
| w VideoChat2 [48] | 51.01 | **2.95** |
| w VideoLLaVA [49] | **51.88** | 2.93 |
| **w Our VGCM** | **62.21** | **3.12** |

reaches an average accuracy of 64.4%, outperforms VGCM without causal inference and VAR by 5.2% and 9.6%, details can be found in Appendix Sec. D.1.

Moreover, to further validate the generalization capabilities of our model, we evaluate the quality of output causal relations on a related and representative video reasoning task: Video Question Answering (VQA) as shown in Tab. 7. Specifically, during inference on the multi-event subset of ActivityNet-QA [59] (The part that overlaps with the MECD test set), we prompted MiniGPT4-video [45] with additional causal relations outputs alongside the standard question inputs. This paradigm facilitates the VLLMs in considering the task from a causal perspective. As shown in the table below, when prompted with these additional causal relations, the answering accuracy of MiniGPT4-video [45] improved by our VGCM surpasses other strong VLLMs like VideoChat2 [48]. These findings confirm that our model can provide accurate causal perception for videos, significantly improving performance on related video reasoning tasks.

**Dataset Robustness** We study the subjectivity and data volume of our proposed MECD dataset, which is shown in Tab. 5. In the experiments of increasing the ratio of randomly flipped annotated causal relations (flipping only one relation of the whole causal relations of video), the accuracy decreases slightly, demonstrating the small amount of subjectivity in labeling does not have a serious impact. Besides, we analyze the scale of data, the increment from 600 examples to 806 examples yields a very modest improvement, indicating the adequacy of our dataset.

## 6  Conclusion

We proposed a novel task, multi-event video causal discovery (MECD), which focuses on event-level causal discovery in long-term videos. Besides, we built the MECD dataset with long-term daily life video datasets with causal relations to support this task and proposed the first video events causal discovery framework VGCM in the principles of Granger Causality. Additionally, our proposed VGCM was facilitated with deeper reasoning ability through causal inference with the chain of thoughts and existence-only descriptions. Our VGCM significantly outperforms GPT-4o and VideoLLaVA by 5.7% and 4.1%, respectively, demonstrating its robust reasoning ability.

# 7 Acknowledgement

The paper is supported in part by the National Natural Science Foundation of China (No. 62325109, U21B2013) and the Lenovo Academic Collaboration Project.

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

# Appendix

## A    Implementation details

**Pretraining process** For each video event, visual features are extracted using ActivityNet pretrained ResNet200 [60], following [10, 37–39]. Prior domain knowledge could benefit the Granger Causality Causal discovery method [61], so we fully pre-trained our model for the dense video captioning task on a 3.1k ActivityNet Captioning video dataset, each video sample contains more than 4 events.

**Training set** All the experiments are conducted on 1 NVIDIA A40 GPU. We train our model for 20 epochs with a learning rate of 16e-5 about 6 hours. Our optimizer is consistent with BertAdam [50] optimizer, with 3 epochs of warm-up. The open-set experiment set can be found in Appendix Sec. D.1. We report the average results during all experiments under three random seeds (2023, 2024, 2025). The ablation of two modalities can be found in Appendix Sec. C.1.

**Model details** Our encoder $Enc_V$, $Enc_C$, and multi-modal video decoder Dec are built upon Videobert [50], a joint model for video and language representation learning. The details of the GPT-4 API prompt can be found in Sec. D.2 in the Appendix.

**Hyperparameters** $\lambda_C$, $\lambda_R$, $\lambda_V$, $\lambda_S$ are set to be 1.0, 4.0, 0.25, 0.05. Maximum input lengths of the caption, the chain of thoughts, and the existence-only descriptions are set to 50.

**Implementation of VAR**[**] We migrate the VAR to our task through an effective method: We mask any event $e_k$, (k<N), and then utilize the fully trained VAR to perform event prediction of $e_k$. If the prediction results $\hat{e_k}$ is obviously various from $e_k$, it is considered that the event $e_k$ is non-causal. Then $r_k$ is labeled as 0; in the opposite case, $r_k$ is labeled as 1. We also report the average results of VAR under three random seeds (2023, 2024, 2025).

**Implementation of LLMs** As for GPT-4 and Gemini-Pro, We report the average results of three calls.

**Implementation of VLLMs** We report the average results of VLLMs under three random seeds (2023, 2024, 2025). When VLLMs do not output $r$ in the required format, we order them to re-answer until the outputs match the format to measure their best performance.

## B    Additional Visualization

### B.1    Successful abduction examples of our VGCM

In Fig. 9, additional examples are presented to showcase the performance of our VGCM, particularly excelling in complex abduction scenarios. The first example successfully discovers that there is no causal relation between *" We see the targets in front of a backdrop."* and *" The instructor walks over to the targets."*, despite the backdrop being a necessary object of the result event. This abduction avoids the illusory existence causality.

The second example successfully discovers that there is no causal relation between *" The video shows different cricket matches taking place where Sri Lanka is playing against teams from different countries."* and *" The stadium is filled with spectators cheering for the cricketers."*, despite the spectators' cheering often happening after the game playing. This abduction avoids the illusory temporal causality. Both instances align with the foundational principles motivating our method design.

The third example shows the 83.3% accuracy of video causal relations abduction. Notably, it correctly discerns the most complex causal relations, however, it fails to realize that person B doesn't hit the tennis ball can contribute to the anticipation of the result event of continuous hitting. This indicates that VGCM might still require refinement in understanding causality within higher-level semantics, especially in the mining of some obscure mental or emotional influences. We will strive to explore further solutions in the follow-up work.

### B.2    Failure abduction examples of GPT-4

While examining the causal discovery results of GPT-4, we encountered some intriguing observations. In the example presented in Fig. 10, the GPT-4 API incorrectly infers that all premise events have a causal relation with the result event of the team winning. However, the initial appearance of the team

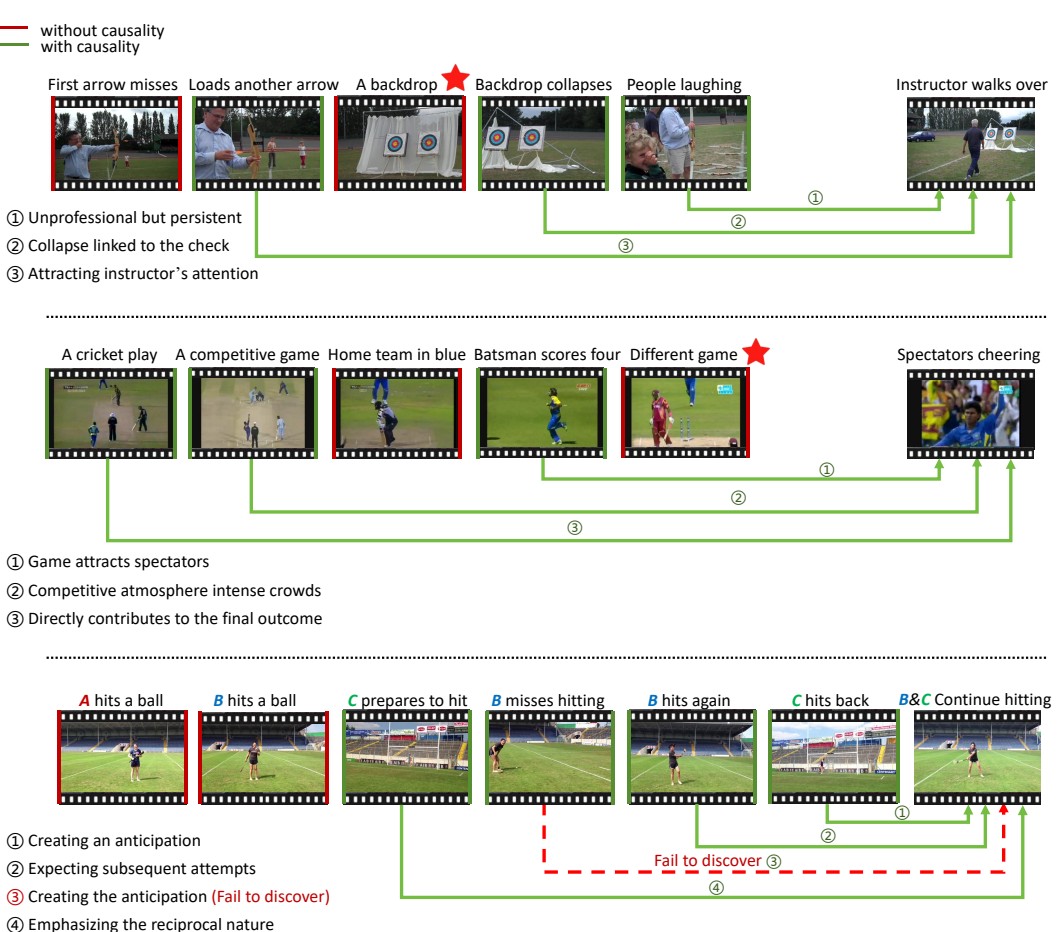

Figure 9: **More successful abduction examples of our proposed VGCM.** The relation which reveals our method of eliminating illusory causality is marked by a red five-pointed star ⭐. The failure case is annotated in a red dotted line ----.

does not directly lead to their victory, and the subsequent celebrations also lack any causal links with the outcome. Indeed, the false discovery by the GPT-4 API could stem from the illusion of causality, where the team's mere presence is perceived as a necessary condition for the outcome. Additionally, the illusion of temporal causality may also play a role, as statistics indicate that celebrations often occur before the announcement of the competition winner. These cognitive biases could contribute to the erroneous causal inference made by the GPT-4 API in this scenario.

When we request a detailed explanation from the GPT-4 API regarding the discovered causal relation between the result event and the initial appearance of the team, the response is *"Setting up the motive for the last event."* Obviously, the GPT-4 confuses causality with the illusion of existence causality. In contrast, our VGCM makes a correct inference in this scenario. Furthermore, when we seek detailed reasons from the GPT-4 API for the discovered causal relations between the result event and the celebrations, the answer is *"Indicating their satisfaction and confidence in their performance, implying they believe they have a good chance to win."* Here, the GPT-4 API misinterprets causality by associating it with the expression of subjective emotions unrelated to the events in question. It may mistake the display of subjective emotions for the presence of objectively implied causality.

## B.3 Annotation pipeline of MECD dataset

To improve the accuracy and mitigate subjective biases in annotating causal relations, we employ a cross-annotation strategy [62–64]. The interactive interface used by annotators during the labeling process of our MECD dataset is illustrated in Fig. 11. Each video example is endowed with a

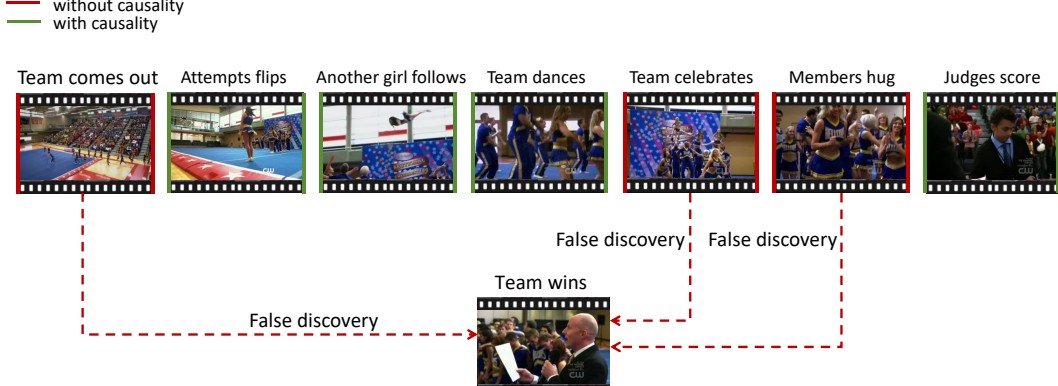

Figure 10: **Failure abduction examples of GPT-4.** Many failure cases of GPT's causal reasoning are due to confusion with illusions and the conflation of subjective emotions with objective laws.

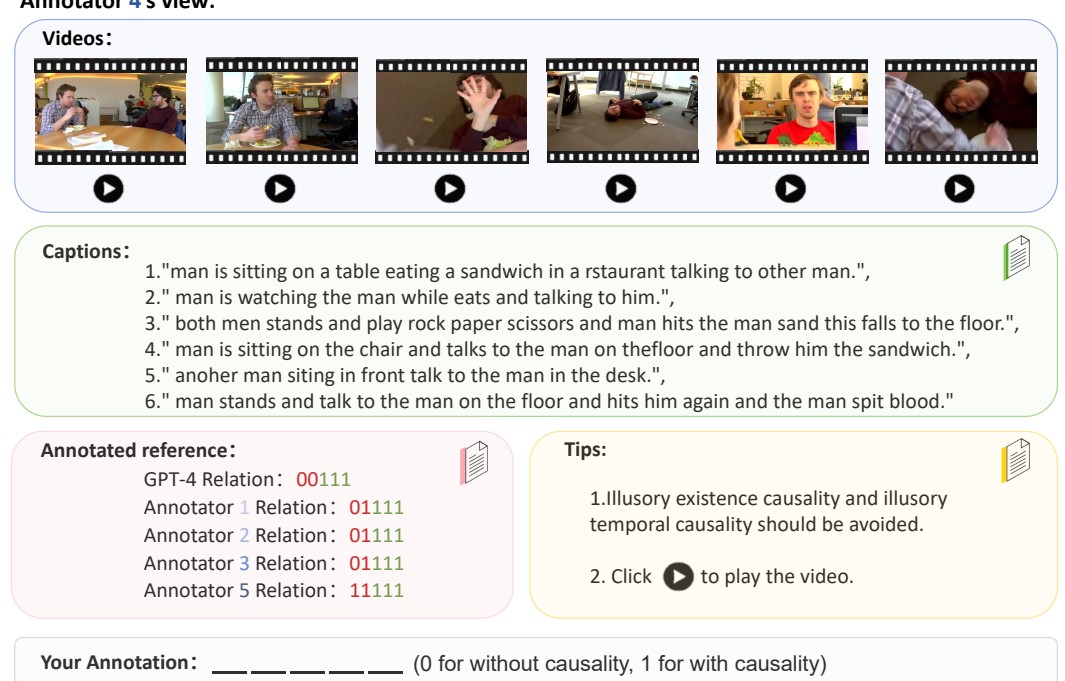

Figure 11: **Annotation pipeline of MECD dataset.** Illustration of the interactive interface used by annotators during the labeling process of our MECD dataset. Key information is provided during annotation.

"*relation*" attribute. First, GPT-4 [33] provides an initial annotation of attribution, which is then further refined by five human annotators. Ground truth labels are determined based on the majority choices of the annotators regarding causal relations. This methodology ensures the creation of a more reliable and objective dataset.

## B.4 Annotation examples of MECD

Annotation examples of MECD are shown in Fig. 12, our MECD dataset is carefully annotated to support the challenging task proposed with complete premise information.

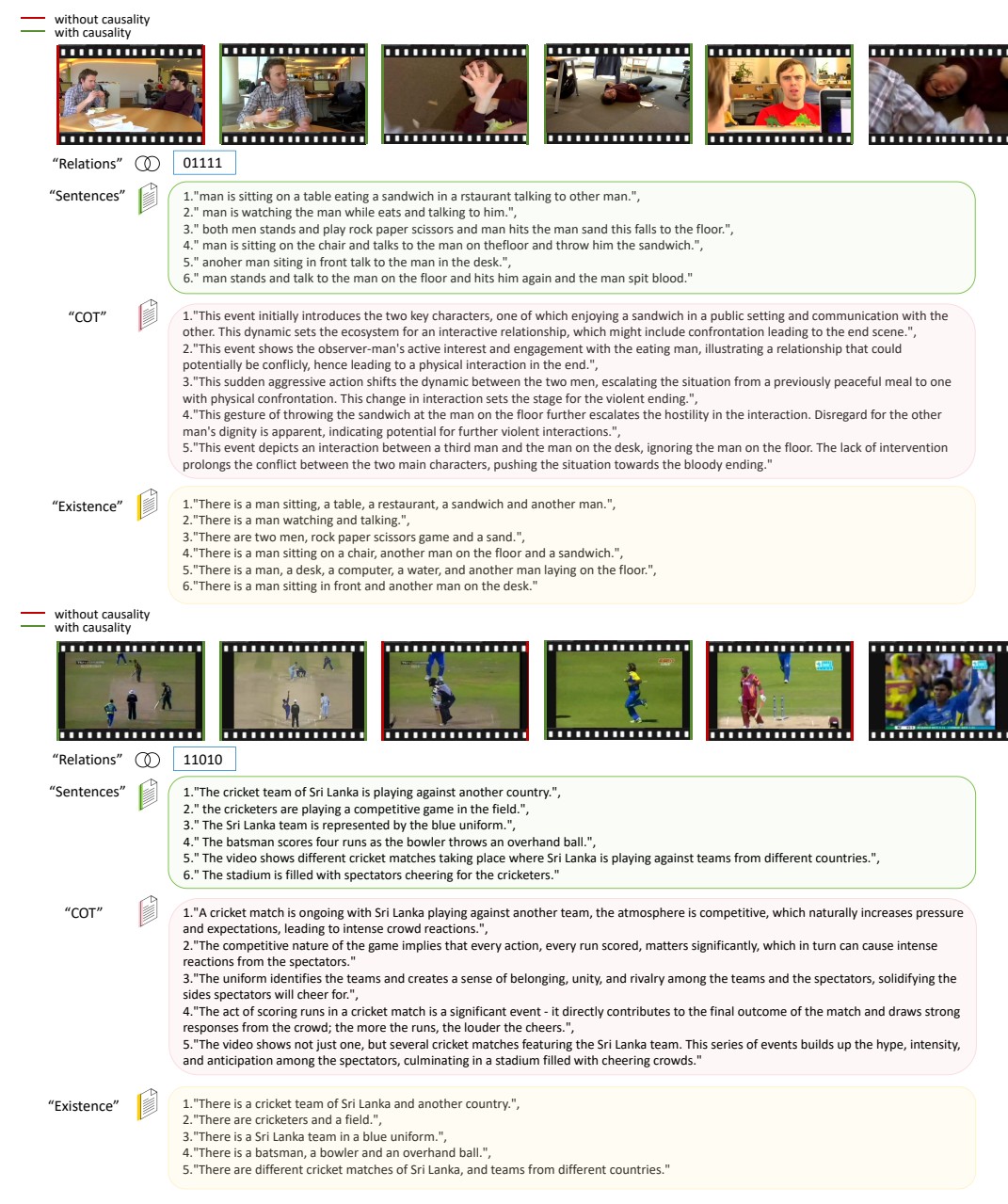

Figure 12: **Annotation examples of MECD.** Annotation examples of MECD are shown. Newly annotated attributes "Relations", "COT", "Existence" and the existing caption attribute "Sentences" are shown along with the video frames.

## C   Additional Experiments

### C.1   Modalities analysis of causality discovering

The MECD task employs both video input and corresponding captions to uncover causality. Our objective in this experiment is to evaluate the degree of reliance on these two modalities in causal discovery. Typically, each event in our MECD task consists of a textual input with an average of 13.5 words caption and a visual input of 50 frames.

To investigate the influence of the text modality, we employ a masking strategy for the input caption of the premise event, gradually increasing the masking ratio from 10% to 80%. The results presented

Table 8: **VGCM performance with masked premise event caption input.** * indicates 30 frames masked at the same time.

| Num of words masked | Accuracy |
| --- | --- |
| non-masked | 71.2 |
| 2 per event | 70.2 |
| 5 per event | 69.7 |
| 8 per event | 69.2 |
| 8 per event* | 67.4 |
| 11 per event | 68.9 |

Table 9: **VGCM performance with masked premise event visual input.** * indicates 10 words masked at the same time.

| Num of frames masked | Accuracy |
| --- | --- |
| non-masked | 71.2 |
| 5 per event | 70.3 |
| 15 per event | 69.0 |
| 20 per event | 68.3 |
| 20 per event* | 67.1 |
| 40 per event | 67.9 |

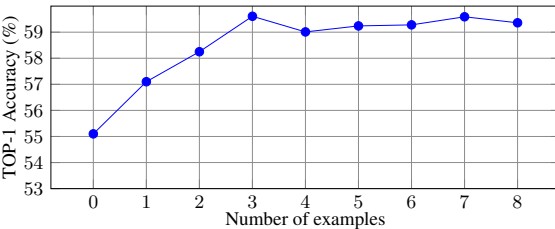

Figure 13: **The trend chart of inference accuracy as the number of examples changes under the In Context Learning paradigm.** Accuracy increases slightly when increasing the number of few-shot examples, when the number of examples > 3, the accuracy tends to remain constant.

in Tab. 8 indicate that our VGCM does not rely on the textual modality input; VGCM can also conduct causal discovery for videos without any captions.

In contrast, the experimental results suggest a more obvious performance decrease towards less visual modality input in the causality discovery task, as shown in Tab. 9. However, even with 80% masking of either modality, the results consistently outperform our strong baseline model, VideoLLaVA, underscoring the robust causal discovery capability of VGCM.

Furthermore, we conducted experiments involving simultaneous masking of both modalities of information. Interestingly, we observe a noticeable decrease in accuracy compared to when only one modality is masked. This observation highlights the importance of jointly considering both modalities in the causality discovery task.

## C.2 Adequacy of the prompts provided to GPT-4

To delve deeper into the limitations of the straightforward baseline approach of prompting GPT-4, we examined the correlation between its accuracy and the number of video examples provided in the few-shot prompts. The findings, illustrated in Fig. 13, suggest that increasing the number of examples shown to GPT-4 does not effectively enhance its accuracy. This suggests that the limitation of the GPT-4 baseline is not strongly correlated with the number of presented examples but rather is more attributable to its intrinsic limitation in understanding complex causal relationships solely through text modality.

# D Experiments details

## D.1 Details of causality discovery experiment

In the open-set experiment of exploring reasoning ability, the five categories mainly consist of the activities below, demonstrating the colorful daily activities included in our dataset.

**Sports:** Arm wrestling, BMX, Beach soccer, Blow-drying hair, Capoeira, Croquet, Futsal, Ice fishing, Kite flying, Playing beach volleyball

**Creating & Making:** Assembling bicycle, Baking cookies, Building sandcastles, Carving jack-o-lanterns, Decorating the Christmas tree, Hanging wallpaper, Making a cake, Making an omelet, Painting fence, Putting in contact lenses

**Daily Activities:** Changing car wheel, Cleaning sink, Drinking coffee, Eating ice cream, Gargling mouthwash, Hanging wallpaper, Kneeling, Peeling potatoes, Putting on shoes, Washing face

**Performing:** Baton twirling, Bullfighting, Drum corps, Fun sliding down, Hula hoop, Playing congas, Playing drums, Playing rubik cube, Playing saxophone, Tumbling

**Socializing:** Beer pong, Playing blackjack, Playing field hockey, Playing harmonica, Playing piano, Playing squash, Playing water polo, Rock climbing, Smoking hookah, Belly dance

### D.2 Prompts to generate auxiliary premise information

In this section, we introduce the detailed method of prompting GPT-4 [33] to generate more premise information. Firstly, we prompt the GPT-4 with the following prompts to generate the description-only sentences.

```
# Task: Each input consists of n sentences, and the text description
of each sentence has been given correspondingly (separated by " ",).
You need to offer the existence description of each sentence.
```

Besides the task description, we further append the few-shot paradigm (In-Context Learning) introduced in [52–54, 65, 66]. Similarly, we prompt the GPT-4 [33] with the following instructions to generate the chain-of-thoughts candidate sentences in the same few-shot paradigm.

```
# Task: Each video consists of n events and the text description
of each event has been given correspondingly separated by " ",).
First n-1 events might be the cause of the last event. You need to
offer the chain of thoughts you derive that causes the last event.
```

### D.3 Chain of thoughts examples

In this section, we present an example of the chain of thoughts prompted, the corresponding premise event and result event descriptions are also shown below:

```
{"premise event sentence": "He continues sharpening the knife, turn it again
to further sharpen the other side and wipe it with paper towel."

"result event sentence": "Throws the old and dirty paper towel and reach the
roll of paper towel and clean the knife."

"COT": "The repeated action of sharpening and wiping the knife  underscores
the importance of both the knife's sharpness and cleaners, leading directly
to the final action of disposing of the used paper towel and getting a new
one to ensure the knife is thoroughly clean"}
```

The chain of thoughts shown above provides a logical causal chain between the event of the cleaning of the knife and the subsequent throwing of the dirty paper towel. The reasoning initiates by considering the heightened need for sharp and pristine knives achieved through sharpening. This causal chain is then expanded by suggesting that this demand could have led to the replacement of the paper towel. The chain of thoughts generated from GPT-4 serves as a candidate in the process of correct reasoning, contributing to the exploration of potential causal relations.

## E   Limitations and future works

1. The video we input for causal discovery needs to provide timestamps, we encourage future work to realize causal discovery with weakly annotated inputs.

2. VGCM might still require refinement in understanding causality within higher-level semantics, especially in the mining of some obscure mental or emotional influences according to the failure cases analysis in Appendix Sec. B.1.

3. VGCM is based on the supervised paradigm of causal discovery, subsequent works may be able to extend to the unsupervised paradigm.

4. The causal graphs proposed by the MECD may also enhance other video understanding tasks, such as video dense captioning and video event prediction, or could be introduced to other reasoning tasks, including text reasoning and mathematical reasoning tasks.

5. The evaluation results of VLLMs and LLMs on the MECD task also help researchers study language models' current issues and limitations in complex reasoning.

