# OpenReview forum: "MECD: Unlocking Multi-Event Causal Discovery in Video Reasoning"
_NeurIPS.cc/2024/Conference — NeurIPS 2024 spotlight_

### Official Review · Reviewer_17Ce · 2024-07-04

**Soundness:** 4
**Presentation:** 4
**Contribution:** 3
**Rating:** 8
**Confidence:** 2

**Summary:**

The paper studies causal reasoning in video, specifically, causal diagrams in long, multi-event videos. To do this, the authors attempt to define MEDC, a new task for discovering the complete causal relation diagram in multi-event chronological videos (~ 2 minutes). For example, analyzing events from traffic surveillance videos across different times, to identify the causes of an accident. Coupled with the task, they create a dataset of 1107 lifestyle videos from ActivityNet with multiple events in them. To facilitate the task, they propose Video Granger Causality Model, a framework inspired by the Granger Causality Method, and show it surpasses existing models (GPT-4, Gemini-1.5 Pro, Video-LLava).

**Strengths:**

* The paper introduces a new, challenging, and important task in video understanding
* The methodology section was interesting to read, with the approach being mostly easy to follow.
* There is a wealth of analyses and ablation work both in the main paper and the supplementary, which cover many aspects of the paper, such as the task itself, the trained model, and the baselines the model is compared to.

**Weaknesses:**

No major weaknesses come to mind.

**Questions:**

N/A

**Limitations:**

yes

---

> ### Author Rebuttal · Authors · 2024-08-07
>
> Thank you for your insightful comments.
>
> We fully agree that the proposed MECD represents a novel, challenging, and significant task in video understanding.
>
> We also appreciate the acknowledgment of the interest and clarity provided by the VGCM method.

---

### Official Review · Reviewer_2Vef · 2024-07-12

**Soundness:** 3
**Presentation:** 3
**Contribution:** 3
**Rating:** 5
**Confidence:** 4

**Summary:**

This paper introduces a new task called Multi-Event Causal Discovery (MECD). Given a video which comprises multiple  temporal events that are chronologically organized,  the goal is to predict if any previous event has a causal effect on the last event in the sequence. The MECD dataset is filtered and curated from the existing ActivityNet v1.3 video dataset. Additionally, the paper also proposes the Video Granger Causality Model (VGCM) approach, which leverages an event prediction model inspired by the Granger Causality method to perform an Event Granger Test. This test estimates causality by comparing predicted result events with masked versus unmasked premise events. Furthermore, the front-door adjustment and counterfactual inference techniques are also incorporated into VCGM. The authors demonstrate the benefits of their proposed VCGM approach by comparing it to state-of-the-art LLMs and VLLMs on the MECD benchmark, where it outperforms the latter by a significant margin.

**Strengths:**

1) Overall, the paper addresses a significant problem in video understanding, especially with the recent interest in understanding the capabilities of visual LLMs (VLLMs). In contrast to how existing visual-language foundation models are pretrained and evaluated, the proposed MECD benchmark and the Video Granger Causality Model are aimed at understanding the causal relationships between multiple temporal events in videos instead of simply describing them.

2) The intuition and theoretical reasoning underlying the proposed  Video Granger Causality Model is sound. In particular, the two main problems of Causality confounding and Illusory Causality in video causal discovery are well-motivated. Additionally, the mathematical formulation for integrating the front-door adjustment and counterfactual inference methods to resolve the listed issues is insightful.

3) The model figures are informative and especially helpful in helping the reader to understand the different stages of the data curation process as well as the intuition behind each stage. The paper is also well-organized and well-written.

**Weaknesses:**

1) While the introduced Multi Events Causal Discovery dataset is a nice contribution as a benchmark, it is relatively limited in size with 1107 data samples. For example, there are only 808 and 299 video samples for training and evaluation, respectively. With such a small number of evaluation samples, it is challenging to evaluate these trained LMMs comprehensively. It may be beneficial to source videos from a wider variety of sources such as long and instructional videos from Youtube. Additionally, why is a validation set not included as part of the splits?

2) The results in Table 1 provide some interesting comparisons between language-only baselines like Gemini 1.5 Pro and VLLM baselines including Minigpt4-video and Video-LLaVA. However, there are some other state-of-the-art video-language models that are not compared to in the paper such as Video-Llama [1]. Additionally, it has been demonstrated before that the image variants of VLLMs actually outperform their video counterparts on some video understanding tasks. It may be beneficial to include comparisons to such models such as InstructBLIP [2] and Llava to make the analysis more comprehensive.

3) In section 4.1, it would be helpful to include a brief overview of the Granger Causality method, which forms the basis of the proposed Video Granger Causality Model.  Additionally, there are some sections, especially those that briefly describe the Event Granger Test and the causal inference techniques, could benefit from additional technical depth and explanations.

4) Another concern lies in the training and evaluation setup. The VGCM approach is built off the VideoBert model and it will be helpful to include the results obtained by the base VideoBert model. Furthermore, the other baselines, that are being compared to, are evaluated under a zero-shot setting. However, the proposed model appears to be trained and evaluated under a strongly-supervised setting with the annotations from the MECD dataset.

[1] Hang Zhang et al. Video-LLaMA: An Instruction-tuned Audio-Visual Language Model for Video Understanding. EMNLP 2023 demo track.

[2] Wenliang Dai et al. InstructBLIP: Towards General-purpose Vision-Language Models with Instruction Tuning. NeurIPS 2023.

**Questions:**

This is a minor note but it will be helpful for readers to define the term ‘premise event’ at the beginning. Also, please look at the above-mentioned limitations.

**Limitations:**

Yes.

---

> ### Author Rebuttal · Authors · 2024-08-07
>
> Thank you for your review and comments. We’re glad to hear that you found our task interesting and that you see our methods as insightful. Please see below for responses to your comments and questions.
>
> **Q1. Dataset Scale**
>
> We have the following points to this issue:
>
> 1. We envision MECD as a long-term, continuously maintained benchmark.  We will progressively expand its scope by incorporating a wider range of video types and increasing data volumes. In the final version, we plan to include an additional validation set as we have collected and annotated more videos.  Nonetheless, it's important to note that the current MECD is already effective in assessing the capabilities of LLMs for video causal discovery.  As demonstrated in Tab. 1 of our main manuscript, even with a limited number of test samples, LLMs struggle to accurately identify and discover causal relations between video events.
>
> 2. In causal discovery tasks, the evaluation scale is more accurately reflected by the number of event pairs rather than just the number of video samples. This is analogous to similar NLP tasks focused on causal discovery within texts or documents. Frequently utilized datasets in this domain, such as SeRI [1-2], Causal-TimeBank [3-4], HiEve [5], and EventStoryLine [6], comprise a limited number of test samples. For instance, Causal-TimeBank, HiEve, and EventStoryLine datasets have only 95, 100, and 318 articles or documents available for testing, respectively. However, as the evaluation is conducted between event pairs within documents, these datasets contain 1,475, 2,282, 2,725, and 4,027 event pairs, respectively. Similarly, our MECD provides 3,681 event pairs during the test phase, ensuring a comprehensive and convincing evaluation.
>
> 3. Finally, as suggested by Reviewer-eXFX, we also try to incorporate additional metrics to provide a more comprehensive evaluation. For example, we newly report the *Ave SHD* metric for different baselines to reveal the ability of complete causal graph discovery for the whole video (cf. our response to Q3 of Reviewer-eXFX).
>
> **Q2. More comparison**
>
> Thank you for your suggestion. In the table below, we included results for Video-Llama and two image-based VLLMs (Instruct BLIP and LLaVA) for a comprehensive comparison. The results demonstrate that image-based LLaVA does not outperform its video-based counterpart Video-LLaVA in our task. This finding further confirms the challenging nature of our task and highlights the necessity of long-term temporal understanding to accurately identify causal relationships between video events.
>
> | LLM          | Acc  |
> | ------------ | ---- |
> | Video-Llama  | 60.6 |
> | InstructBLIP | 59.7 |
> | LLaVA        | 60.2 |
> | Video-LLaVA  | 62.5 |
>
> **Q3. Overview of causal concepts**
>
> Thanks for your valuable advice. We will add an overview of the causal concepts to the final version.
>
> **Q4-1. Base VideoBERT performance**
>
> We have also conducted experiments on our baseline model VideoBERT, achieving an accuracy of 60.9%. Our VGCM model (71.2%) has made a significant improvement compared to the baseline model.
>
> **Q4-2. Training and evaluation under strongly-supervised setting**
>
> For base multi-modal models VAR and Videobert, we report results from models fully trained on the MECD dataset. All LLM-based models are evaluated under a few-shot setting rather than zero-shot (cf. supplementary Sec. I, lines 600-606). Specifically, following the approach in causal discovery for NLP tasks [2, 7], three representative examples are provided during inference. We also investigated the impact of varying the number of few-shot examples, demonstrating the adequacy of our prompt (cf. supplementary Sec. G, lines 572-578).
>
> Furthermore, as you suggested, to ensure a fairer comparison, we conducted a strongly-supervised experiment on the advanced VLLM method Video-LLaVA. Specifically, we fine-tuned its LLM and encoder components using LoRA under its official implementation on our entire MECD training set. As shown in the table below, Video-LLaVA gains a 4.6% improvement from fine-tuning on our dataset. However, it still falls behind our proposed method. Given that fine-tuning LLM-based baselines is time-consuming, we will include more results of VLLMs with strongly-supervised settings for a comprehensive comparison in our final version.
>
> | Setting                  | Acc      |
> | ------------------------ | -------- |
> | Video-LLaVA (few-shot)   | 62.5     |
> | Video-LLaVA (fine-tuned) | 67.1     |
> | Ours                     | **71.2** |
>
> **Q5. Definition of premise event**
>
> Thanks for your advice, the premise event is the event that happens earlier than the result event in the same video, and we will add the definition to the final version.
>
> **References**
>
> [1] Seri: A dataset for sub-event relation inference from an encyclopedia. NLPCC 2018.
>
> [2] Reasoning subevent relation over heterogeneous event graph. KIS 2024.
>
> [3] Annotating causality in the TempEval-3 corpus. EACL 2014.
>
> [4] Learning to teach large language models logical reasoning. arXiv 2023.
>
> [5] HiEve: A corpus for extracting event hierarchies from news stories. LREC 2014.
>
> [6] Neural granger causality. TPAMI 2021.
>
> [7] Is chatgpt a good causal reasoner? a comprehensive evaluation. EMNLP 2023.

---

> > ### Comment · Reviewer_2Vef · 2024-08-13
> >
> > Thank you very much for your detailed responses to my questions. In particular, I find the addition of the new Ave SHD metric to be helpful and especially insightful for understanding the limitations of existing large multimodal models. Consequently, I will retain my initial rating.

---

> > > ### Author Response · Authors · 2024-08-13
> > > **Response to reviewer-2Vef**
> > >
> > > Thank you for your response.
> > >
> > > We appreciate your acknowledgment of the usefulness of our rebuttal. We confirm that we will include the additional SHD metric and improve the comparison methods as you suggested in the final version of the manuscript.
> > >
> > >
> > > Once again, we are grateful for your valuable feedback and acknowledgment of our work.

---

### Official Review · Reviewer_eXFX · 2024-07-12

**Soundness:** 3
**Presentation:** 3
**Contribution:** 3
**Rating:** 5
**Confidence:** 4

**Summary:**

The paper introduces a new task and dataset, Multi-Event Causal Discovery (MECD) to better understand long videos in a casual perspective. Inspired by the Granger Causality method, the authors devise a framework, dubbed VGCM, to perform the Event Granger Test. Also, VGCM is combined with front-door adjustment and counterfactual inference to tackle the issues of causality confounding and illusory causality. Experiments show that VGCM outperforms GPT-4 and Video-LLaVA in providing causal relationships in multi-event videos.

**Strengths:**

- To advance comprehensive and structured causality analysis for videos with multiple events, the authors introduce a benchmark for the multi-event causal discovery task.
- The authors have crafted an innovative model framework that integrates the Event Granger Test with various causal inference techniques.
- Experiments demonstrate the efficacy of the proposed framework in providing causal relationships within multi-event videos.

**Weaknesses:**

- To enhance the demonstration of the model's generalizability, it is recommended to conduct experiments on a variety of related datasets.
- In Figures 1(c) and 1(e), the events occurring within the video frames are not readily discernible. Additional verbal descriptions are recommended to facilitate a clearer understanding of the figures' intended message.

**Questions:**

- The authors employ Accuracy as the primary performance metric. Could the authors consider or design some other metrics that are more suitable for Multi-Event Causal Discovery (MECD)?
- It would be highly beneficial if the dataset and associated code could be made publicly available at the earliest opportunity.
- Regarding lines 154-155, I suspect that the confounding factor in the spurious causal relationship could introduce significant discrepancies when comparing the output features with and without the factor. Could the authors provide further analysis on this matter?

**Limitations:**

The model depends largely on video captions, which makes it possibly incapable of processing video datasets without captions or descriptions.

---

> ### Author Rebuttal · Authors · 2024-08-07
>
> Thanks for your suggestions and for recognizing the novelty and contribution of our work. Please see the responses to your comments.
>
> **Q1. Model's generalizability**
>
> To the best of our knowledge, our benchmark is currently the first and only one for the video causal discovery task. To further validate the generalization capabilities of our model, we evaluated the quality of output causal relations on a related and representative video reasoning task: Video Question Answering (VQA).
>
> Specifically, during inference on the ActivityNet-QA dataset, we prompted Minigpt4-video with additional causal relations outputs alongside the standard question inputs. This paradigm facilitates the VLLMs to consider the task from a causal perspective. As shown in the table below, when prompted with these additional causal relations, the answering accuracy of Minigpt4-video improved by our VGCM surpasses other strong VLLMs like VideoChat2. These findings confirm that our model can provide accurate causal perception for videos, significantly improving performance on related video reasoning tasks.
>
> | Output Causal Relations             | &nbsp;&nbsp;VQA Acc | VQA Score |
> | ----------------------------------- | :-----------------: | :-------: |
> | w/o (Standard QA setting for VLLMs) |        43.17        |   2.82    |
> | w Gemini-Pro                        |        49.10        |   2.90    |
> | w GPT-4                             |        49.36        |   2.89    |
> | w VideoChat2                        |        51.01        | **2.95**  |
> | w VideoLLaVA                        |      **51.88**      |   2.93    |
> | **w Our VGCM**                      |      **62.21**      | **3.12**  |
>
> **Q2. Improve the display of Fig.1**
>
> Thanks for your advice, in the final version, we will appropriately explain the key elements in Fig.1 and the revised version of Fig.1 can be found in the attached PDF file in our rebuttal.
>
> **Q3. Additional metrics**
>
> As mentioned in our main manuscript (Sec. 2 lines 100-104 & Sec. 5.4 lines 318-321), our model can output a complete causal diagram, consequently, we can introduce Structural Hamming Distance (SHD) [1-2] as a supplementary metric. SHD measures the degree of matching between causal graphs by summing the number of incorrect causal relations. Compared to the Acc metric, it focus more on the global causal relationships for all events in each video.
>
> In our MECD test set, the average number of causal relations in video causal graphs is 12.31. We report the average SHD for incomprehension in the following table, a lower Ave SHD value indicates better performance.
>
> | | Models | Ave SHD (12.31) $\downarrow$ | Acc $\uparrow$ |
> | :------------------: | ------------------- | :--------------------------: | :------------: |
> | LLM | Gemini-1.5-Pro      |             4.91             |      59.3      |
> | | GPT-4 |             4.92             |      59.6      |
> | Video LLM | Minigpt4-Video      |             5.16             |      56.8      |
> |  | Minigpt-4 |             5.14             |      57.5      |
> | | VideoChat2 |             4.89             |      60.7      |
> | | VideoLLaVA |           **4.85**           |    **62.5**    |
> | Multi-Modal Backbone | VAR                 |             4.96             |      57.3      |
> | | Videobert  |             4.95             |      60.9      |
> | **Ours** | **VGCM(Videobert)** |           **4.19**           |    **71.2**    |
>
> The results also indicate that for a majority of the models, the current metric, accuracy in identifying causal relations leading to the result event, is already adequate to represent their causal discovery capabilities. However, Gemini and GPT-4 exhibit a superior overall capacity for discovering complete causal relations.
>
> **Q4. Dataset & Code available**
>
> Thanks for your suggestion. We confirm that our code and dataset will be released as soon as possible.
>
> **Q5. Confounding factor**
>
> As you have thought, the confounding factor is unfavorable in causal relation discovery. For a further analysis of lines 205-226: When $e_k$ is masked for comparison, the causal relations between $e_k$'s adjacent events and the last event $e_N$ are affected, leading to a confounding of causal effects.
>
> In lines 154-155, we conduct the Event Causality Test to compare the predictions of the two streams. Consequently, under the concept of control variables, as we want to evaluate the causal effect between the current event $e_k$ and $e_N$, we must prevent the causal effects between the adjacent events and $e_N$ from being affected by confounding factors. Confounding factors result in a redundant or missing causal effect. Therefore, we introduce causal inference to cut off the redundant causal effect between $e_{k+1}$ and $e_N$ by counterfactual intervention. Likewise, by reconstructing $e_k$'s causal effect through front-door adjustment, we introduce the distinction after reconstruction in the causal effect between $e_{k-1}$ and $e_N$ as the compensation.
>
> **Q6. Limitation: Captions dependency**
>
> Currently, with the rapid development of VLLMs, we believe that generating accurate captions for videos will be much easier.
>
> Moreover, in our supplementary (Sec. E, lines 554-567), we have experimented with the dependency of the caption and video inputs, respectively. Specifically, we explored the effect of altering the input format of the two modalities and found that when we masked out 80% of the captions, the accuracy only dropped by 2.3%, proving that we can also conduct tests for videos without any captions.
>
> Thanks for pointing this and we'll delve deeper into this issue in our future work.
>
> **References**
>
> [1] Knowledge transfer for causal discovery. IJAR 2022.
>
> [2] Tuning causal discovery algorithms. PMLR 2020.

---

> > ### Comment · Reviewer_eXFX · 2024-08-12
> >
> > Thank you for your detailed response. My concerns have been addressed, and after considering the feedback from other reviewers, I will maintain my current rating.

---

> > > ### Author Response · Authors · 2024-08-12
> > >
> > > Thanks for your response.
> > >
> > > We greatly appreciate your acknowledgment that our rebuttal has effectively addressed your concerns. We have duly noted your suggestion regarding the model's generalizability and the additional metric in the final version of the manuscript. We will ensure that this change is made to improve our work.
> > >
> > > Once again, we would like to express our gratitude for your valuable feedback and for contributing to the improvement of our manuscript.

---

### Official Review · Reviewer_9my4 · 2024-07-13

**Soundness:** 3
**Presentation:** 2
**Contribution:** 3
**Rating:** 5
**Confidence:** 4

**Summary:**

The paper introduces the Multi-Event Causal Discovery (MECD) task, aiming to uncover causal relationships in videos with multiple events. It presents a novel framework inspired by the Granger Causality method, utilizing a mask-based event prediction model to perform causal inference. The paper also introduces a new dataset for training and evaluation, demonstrating the framework's effectiveness in outperforming existing models like GPT-4 and Video-LLaVA.

**Strengths:**

- Novelty and Relevance: The paper addresses a significant gap in video reasoning tasks by focusing on multi-event causal discovery, which is more reflective of real-world scenarios.
- Framework Design: The use of Granger Causality, combined with advanced causal inference techniques like front-door adjustment and counterfactual inference, is innovative and well-justified.
- Empirical Validation: The framework demonstrates substantial performance improvements over state-of-the-art models, indicating the robustness and efficacy of the proposed approach.
- Dataset Contribution: The creation of the MECD dataset, with detailed annotations and diverse scenarios, is a valuable contribution to the field.

**Weaknesses:**

- Complexity of Implementation: The framework's reliance on multiple advanced techniques may pose challenges for implementation and replication by other researchers. Besides, does VGCM run slower than those VLLM baselines?
- Missing Details: What's the architecture of different encoders? How do the authors pretrain them? How do different loss design? The authors should give more details about fair comparisons.
- Some Typos:
  - In Figure 3, the box colors for the masked and unmasked input should be green and orange.
  - In Line 166, it should be $Env_{V}$ and $Env_{C}$.

**Questions:**

I appreciate that the authors proposed a more novel and interesting setting. However, given the missing details and overly complicated pipeline, I tend to reject this paper for now. And I hope the authors can provide more details about the training and architecture. This will help us reviewers to make a fair evaluation.

**Limitations:**

Yes.

---

> ### Author Rebuttal · Authors · 2024-08-07
>
> Thanks for your valuable suggestions. We’re glad that you found our work interesting and novel. We address your concerns below.
>
> **Q1. Complexity of implementation**
>
> We have the following points for this question:
>
> 1. *The main pipeline of our framework is clear and not overly complex in its implementation*:
>
> - Our basic pipeline (Sec. 4.1) follows a straightforward dual-branch architecture for processing visual and textual inputs. Semantic and causal information interact between two branches, culminating in causality estimation via a masked-based prediction task in both modalities.
>
> - While the causal inference component (Sec. 4.2) may require some background knowledge to fully grasp theoretically, its practical implementation is neat and clear. In our final manuscript, we will provide pseudocode to further clarify these aspects.
>
> - Contrary to the reviewer's concerns, our framework does not rely on numerous advanced techniques or require additional large-scale pretraining. It only requires pretrained visual and textual feature encoders, which are fundamental and readily accessible in the multi-modality understanding community.
>
> - Our framework is also highly flexible, allowing researchers to replace or modify specific modules for targeted improvements. Options include selecting different visual/textual encoders, altering interaction methods between modalities, or adjusting the similarity evaluation criteria for the prediction task.
>
> 2. *The simplicity of our implementation is further evidenced by its parameters and inference speed*:
>
> - Our model contains only 144M parameters, significantly smaller than even the lightest VLLM models (e.g., Video-LLaVA, VideoChat2), which have 7B parameters (~50 times more than ours).
>
> - Our method is fast (0.76s/sample) in inference speed. The overall method introduces only 8.57% overheads to the VideoBERT baseline. Notably, our inference speed is **3-6 times faster** than all VLLMs (as shown in Table 1). Our inference speed experiment is conducted on 1 NVIDIA A6000 GPU.
>
> 3. *To facilitate replication and further research, we will release our code along with the dataset as soon as possible. This will enable other researchers to reproduce our results and build upon our work*.
>
> | Model| Inference Speed (seconds/sample) |
> | ----------------------------------- | :-------------------: |
> | VideoBERT|0.70 |
> | *Our VGCM ( built upon VideoBERT )* |**0.76**|
> | Video-LLaVA|2.12|
> | VideoChat2|2.96|
> | Minigpt4-Video|3.98|
> | Minigpt4|4.72|
>
> *[Table 1. Efficiency of different models. Reported with average inference speed of each sample.]*
>
> **Q2-1. Details of encoders**
>
> We do have provided more details about the architecture of encoders in our supplementary (Sec. B lines 510-511). Specifically, our encoder  $\text{Enc}_v$ and $\text{Enc}_c$ are built upon VideoBERT, a joint model for video and language representation learning.
>
> Besides, as also provided in our supplementary (Sec. B lines 501-502), the architecture of our pretrained encoder $\Phi_{pre}$ is ResNet-200, following the setting of the Dense Video Captioning task [1-2].
>
> **Q2-2. Details of pretraining**
>
> We have provided the detailed pre-training procedures in our supplementary (Sec. B, lines 501-504). For $\text{Enc}_v$ and $\text{Enc}_c$, our pre-training approach is relatively modest. We conducted a video dense captioning task on 3.1K samples from the ActivityNet Caption dataset to warm up the model. It requires a low resource footprint and time cost (2 NVIDIA A6000 GPUs for 8 hours), while leading to a significant improvement in final performance from 69.5 to 71.2.
>
> **Q2-3. Details of loss design**
>
> Overall, the causal relation loss $L_R$ provides direct supervision for output causal relations using standard Cross-Entropy loss, while $L_C$, $L_V$, and $L_S$ aim to further enhance the causal discovery capability from different aspects.
>
> Specifically, $L_C$ and $L_V$ introduce auxiliary caption and reconstruction losses to facilitate event prediction, implemented by Label Smoothing loss and MSE loss respectively. They are incorporated because the Granger Causality Method determines whether an earlier event aids in predicting a subsequent one.
>
> $L_S$, the similarity loss (implemented by InfoNCE), prompting causal relation discovery by comparing output causal feature similarity when a premise event is masked. The rationale behind this loss is that masking a non-causal event should result in a prediction of the result event similar to that of the unmasked stream.
>
> In our manuscript, we have thoroughly examined the contribution of each loss term in Tab. 2 and Sec. 5.2 (lines 285-289). Furthermore, we have detailed the balance between these losses in Sec. 4.1 (lines 188-200), and outlined the specific hyperparameter settings in our supplementary (Sec. B lines 513-514).
>
> **Q2-4. Details about the training**
>
> Details about the training procedures have been provided in our supplementary (cf. Sec. B, lines 505-509).
>
> **Q2-5. Details about fair comparisons**
>
> To make a fair comparison, for multi-modal backbones VAR and VideoBERT, we reported their results after being fully trained on the MECD dataset. For all LLM-based models, we reported the results under the few-shot paradigm (cf. supplementary Sec. I, lines 601-606), where three representative examples are provided during inference, similar to the paradigm used in causal discovery in NLP tasks [3-4].
>
> **Q3. Typos**
>
> Thank you for your corrections. We will correct them in our final version.
>
> **References**
>
> [1] Visual abductive reasoning. CVPR 2022.
>
> [2] End-to-end dense video captioning with parallel decoding. CVPR 2021.
>
> [3] Reasoning subevent relation over heterogeneous event graph. KIS 2024.
>
> [4] Is chatgpt a good causal reasoner? a comprehensive evaluation. EMNLP 2023.

---

> > ### Comment · Reviewer_9my4 · 2024-08-13
> > **Response to authors**
> >
> > The authors solve part of my issues. However, some key issues are not clear, like the details to reproduce the whole results. It will be better if the authors can provide detailed training hyperparameters, architecture, data in clear tables, and even the code.
> >
> > Besides, the LLM-based models are not fine-tuned with the proposed MECD, which is unfair. Can the authors provide more fair results with training on VideoChat or Video-LLaVA. And the authors should clearly claim they use `few-shot paradigm` in the main text.
> >
> > Considering the problems, I maintain my original rating.

---

> ### Author Response · Authors · 2024-08-13
> **Response to reviewer-9my4**
>
> Thanks for your great efforts and time in reviewing our paper.
>
> ### 1. *Results of fine-tuned Video-LLaVA*
>
> Firstly, we want to state that the few-shot evaluation (In-Context Learning) for LLMs is widely recognized as a strong baseline for reasoning and causal discovery tasks [1-6], effectively reflecting their performance on downstream tasks. Therefore, comparing LLMs in a few-shot setting is a common and accepted practice in the field.
>
> Secondly, neither GPT-4 nor Gemini-Pro offer interfaces for fine-tuning, consequently, we reported the few-shot results of LLMs and investigated the impact of varying the number of few-shot examples, demonstrating the adequacy of our prompt (cf. supplementary Sec. G, lines 572-578).
>
> Furthermore, as you and reviewer-2Vef suggested, to ensure a fairer comparison, we conducted a strongly-supervised experiment on the open-source method Video-LLaVA. Specifically, we fine-tuned Video-LLaVA using LoRA under its official implementation on our MECD training set. As shown in the table, Video-LLaVA gains a 4.6% improvement from fine-tuning on our MECD. **However, it still falls behind our proposed method**. Since fine-tuning LLM-based baselines is time-consuming, we will include more results of fine-tuning VLLMs for a comprehensive comparison in our final version.
>
> |Setting| Acc|
> |------------------------ | -------- |
> |Video-LLaVA (few-shot)| 62.5|
> |Video-LLaVA (fine-tuned) |67.1|
> |Ours|**71.2**|
>
> [1] Language models are few-shot learners. NeurIPS 2020.
>
> [2] Rethinking the role of demonstrations: what makes in-context learning work? EMNLP 2022.
>
> [3] Large language models are latent variable models: Explaining and finding good demonstrations for in-context learning. NeurIPS 2023.
>
> [4] CLADDER: assessing causal reasoning in language models. NeurIPS 2023.
>
> [5] Causal inference using llm-guided discovery. AAAI 2024.
>
> [6] Open Event Causality Extraction by the Assistance of LLM in Task Annotation, Dataset, and Method. ACL 2024.
>
>
> ### 2. *More details*
>
> We're happy to provide more details here to clarify your concerns. However, we need to state that **these details can be also found in the supplementary in our initial submission**. To further relieve your concerns about our implementation and reproducibility, we provide an **anonymous GitHub project** (https://anonymous.4open.science/r/NeurIPS-4887-MECD), which contains our *data annotation*,  *training and evaluation codes*, and details about how we evaluate LLMs and how we fine-tuning the Video-LLaVA. Additional details can be found in the README.md file.
>
> - **Training procedure**: All the experiments are conducted on 1 NVIDIA A40 GPU. We train our model for 20 epochs with a learning rate of 16e-5 about 6 hours. Our optimizer is consistent with the BertAdam optimizer, with 3 epochs of warm-up. We report the average results during all experiments under three random seeds (2023, 2024, 2025).
> - **Hyperparameters**: $\lambda_C$, $\lambda_R$, $\lambda_V$,$\lambda_S$ are set to be $1.0, 4.0, 0.25, 0.05$.
> - **Encoder & Decoder architecture**: Our encoder $Enc_{v}$, $Enc_{c}$, and multi-modal video decoder $Dec$ are built upon videoBERT, a joint model for video and language representation learning.
> - **Evaluation of LLMs**: We prompt the GPT-4 with the following few-shot prompts to conduct evaluation (Details can be found in the anonymous GitHub project):
>
> ```
> # Task: Each video consists of n events, and the text description of each event has been given correspondingly (separated by " ",). You need to judge whether the first n-1 events in the video are the cause of the last event, and the probability of the cause 0(uncausal) or 1(causal) is expressed as the output, Let's think step by step through the chain of thought.
> Here are several examples of judging whether the first n-1 events in the video are the cause of the last event:
> <start>
> First example:
> Text description of n events:
> ...
> The probability output should be:
> ...
>
> Second example:
> Text description of n events:
> ...
> The probability output should be:
> ...
>
> Third example:
> Text description of n events:
> ...
> The probability output should be:
> ...
> <end>
> ```
>
>
> As a long-term maintenance benchmark, we confirm that the dataset and code will be public after being accepted.
>
> We hope our responses can address your concerns and raise your rating of our paper.

---

> > ### Comment · Reviewer_9my4 · 2024-08-14
> > **Response to authors**
> >
> > Thanks for the further claim, and I slightly raise the rating. I suggest adding these additional results and details in the final version to make the whole paper more reliable and reproducible.

---

> > > ### Author Response · Authors · 2024-08-14
> > > **Response to reviewer-9my4**
> > >
> > > Thanks for your response.
> > >
> > > We really appreciate your acknowledgment that our further claim effectively addressed your concerns. We will provide the additional results and all the details in the final version of the manuscript following your suggestion. Besides, we will make the complete dataset and codes available as soon as the manuscript is accepted.
> > >
> > > Once again, we would like to express our sincere gratitude for your positive feedback and for contributing to our manuscript.

---

### Author Rebuttal · Authors · 2024-08-07

We sincerely appreciate all reviewers’ time and efforts in reviewing our paper. We are glad to find that reviewers generally recognized our contributions:

- Novelty and contribution of MECD task (Reviewer-9my4, eXFX, 2Vef, 17Ce)
- Innovative and well-motivated framework design of VGCM (Reviewer-9my4, eXFX, 2Vef, 17Ce)
- Strong experimental results (Reviewer-9my4, eXFX, 17Ce)
- Well-organized presentation, clear description (Reviewer-2Vef, 17Ce)

Regarding the concerns proposed by reviewers, our main responses can be summarized as follows:

- **More details about our framework and training** (cf. response to Q2 of Reviewer-9my4)

- **Model's generalizability on related VQA task** (cf. response to Q1 of Reviewer-eXFX)
- **New evaluation metric** (cf. response to Q3 of Reviewer-eXFX)
- **Results of VLLM under strongly-supervised setting in our dataset** (cf. response to Q4 of Reviewer-2Vef)

All the reviewers can refer to the attached PDF for the task definition (Fig.1, Reviewer-eXFX) and a clear display of our pipeline (Fig.2, Reviewer-9my4).

We hope our responses could address your concerns. Please let us know if any clarification or additional experiments would further strengthen the paper. We would be happy to incorporate all these suggestions in the final version.

---

### Author Response · Authors · 2024-08-12
**Kind Request for Discussions and Feedback for Paper 4887**

Dear Reviewers,

Firstly, we'd like to express our sincere gratitude for taking the time to review our paper and for providing invaluable feedback.

Since the deadline for discussion is just around the corner, if you still have any concerns about our paper after reading our response, please feel free to contact us by adding comments.

Your engagement during this discussion period is crucial for the improvement of our work. We genuinely value your insights and look forward to your continued feedback.

Thank you for your time and consideration.

Best regards,

Authors of Paper 4887

---

### Decision · Program_Chairs · 2024-09-25

**Decision:**

Accept (spotlight)

**Comment:**

This submission targets the problem of causal relationships from video with multiple events. The authors proposed a method with a mask-based event prediction model and to capture causal inference. This submission is well-written, and easy to follow. All the reviews give positive ratings to this submission. The AC recommend to accept this submission as a spotlight paper.